# Gene Structure, Expression and Function Analysis of *MEF2* in the Pacific White Shrimp *Litopenaeus vannamei*

**DOI:** 10.3390/ijms24065832

**Published:** 2023-03-18

**Authors:** Yanting Xia, Xiaoyun Zhong, Xiaoxi Zhang, Xiaojun Zhang, Jianbo Yuan, Chengzhang Liu, Zhenxia Sha, Fuhua Li

**Affiliations:** 1School of Life and Sciences, Qingdao University, Qingdao 266071, China; 2CAS and Shandong Province Key Laboratory of Experimental Marine Biology, Institute of Oceanology, Chinese Academy of Sciences, Qingdao 266071, China; 3College of Earth Science, University of Chinese Academy of Sciences, Beijing 100049, China; 4Laboratory for Marine Biology and Biotechnology, Qingdao National Laboratory for Marine Science and Technology, Qingdao 266237, China; 5Center for Ocean Mega-Science, Chinese Academy of Sciences, Qingdao 266071, China

**Keywords:** *MEF2* gene, *Litopenaeus vannamei*, splice variants, expression, growth, immunity

## Abstract

The Pacific white shrimp *Litopenaeus vannamei* is the most economically important crustacean in the world. The growth and development of shrimp muscle has always been the focus of attention. Myocyte Enhancer Factor 2 (MEF2), a member of MADS transcription factor, has an essential influence on various growth and development programs, including myogenesis. In this study, based on the genome and transcriptome data of *L*. *vannamei*, the gene structure and expression profiles of *MEF2* were characterized. We found that the *LvMEF2* was widely expressed in various tissues, mainly in the Oka organ, brain, intestine, heart, and muscle. Moreover, *LvMEF2* has a large number of splice variants, and the main forms are the mutually exclusive exon and alternative 5′ splice site. The expression profiles of the *LvMEF2* splice variants varied under different conditions. Interestingly, some splice variants have tissue or developmental expression specificity. After RNA interference into *LvMEF2*, the increment in the body length and weight decreased significantly and even caused death, suggesting that *LvMEF2* can affect the growth and survival of *L. vannamei*. Transcriptome analysis showed that after *LvMEF2* was knocked down, the protein synthesis and immune-related pathways were affected, and the associated muscle protein synthesis decreased, indicating that *LvMEF2* affected muscle formation and the immune system. The results provide an important basis for future studies of the *MEF2* gene and the mechanism of muscle growth and development in shrimp.

## 1. Introduction

The abdominal muscles of penaeid shrimps account for more than 65% of their body weight. Improving muscle production and quality is an important topic in the shrimp industry. The muscle type of the shrimp is primarily striated muscle [1,2], and a series of intricate morphological, physiological and biochemical changes occur during the growth and development process of shrimp muscles. However, there are few studies on the genes regulating muscle growth and development in shrimp.

Myocyte Enhancer Factor 2 (MEF2), a member of the MADS transcription factor family [3], was first discovered in mammalian muscle cell culture, which directs various cellular functions, including muscle cell development and growth [4]. There are four paralogs in invertebrates, designated MEF2A, B, C, and D [5]; meanwhile, in invertebrates, such as sea urchins and *Drosophila*, there is only single *MEF2* gene. The *MEF2* gene of vertebrates has various forms of alternative splice variants; for example, in common carp (*Cyprinus carpio*) the invertebrate subfamilies, except *MEF2B*, all have alternative splicing, among which *MEF2C* has the most splice variants [6]; in chimpanzees (*Pan troglodytes*), 16 and 17 transcripts were found for the *MEF2C* and *MEF2D* genes, respectively [7]. Numerous isoforms and complex alternative splicing result in structural diversity in the vertebrate *MEF2* gene.

Structural diversity corresponds to functional complexity. The critical functional domains of *MEF2* include the N-terminal highly conserved MADS-box (determining DNA-binding specificity) and the *MEF2* domain (for high-affinity DNA binding and dimerization) [8]; meanwhile, its C-terminal, which is required for transcriptional activation, varies widely among species. In addition, *MEF2* also has a conserved domain HJURP-C, whose function remains to be analyzed. The *MEF2* is a member of the evolutionarily ancient MADS family of transcription factors [9], and the active site of *MEF2* has been found to exist in the cis-regulatory modules (CRMs) of muscle-specific genes, which regulate muscle gene expression [10]. In livestock, *MEF2* could coordinately control the type of muscle fiber differentiation by binding in the E-box region of the *MSTN* promoter region. *MSTN* promoter activity is significantly enhanced by *MEF2C* overexpression [11]. The 5’UTR (untranslated region) of sheep myosin light-chain family genes contains the binding sites of *MEF2*, which indicates that the transcription factor is important for muscle growth and contraction [12,13]. It had been found that MRFS family members (MyoD, MyoG, MRF5, and MRF4) and MEF2 could act synergistically to not only promote, but also inhibit skeletal muscle differentiation. For example, a study in rabbits showed that TRIM72, a novel negative feedback regulator of muscle, was co-activated at the transcriptional level by *MyoD* (or *myogenin*) and *MEF2*, which bind to the E-box and *MEF2* sites of the *TRIM72* promoter, respectively. Mutations in these sites resulted in decreased *TRIM72* promoter activity; however, the activity could be restored by synergistically enhancing *MyoD* and *MEF2* [14]. Recent studies have also found that MRF4 can control muscle mass by inhibiting the activity of *MEF2* in adult skeletal muscle [15]. Other studies have shown that *MEF2* also plays an important role in neurodevelopment and cancer development [16]. In carp, *MEF2A*, *MEF2B*, and *MEF2Cb* play a role in early neurodevelopment, and *MEF2A*, *MEF2Ca*, *MEF2Cb*, and *MEF2D* play a role in early carp muscle development [6]. However, the structure and functions of crustacean *MEF2* genes remain largely unexplored.

Previous studies on crustacean muscle growth-related genes have mainly focused on *Actin* [17], *Myosin* [18], *myostatin* (*Mstn*) [19], *p38* [20], and so on, while the *MEF2* gene has barely been investigated. In 2015, the *MEF2* gene was first identified in the mesoderm of the kuruma shrimp *Marsupenaeus japonicus* [21]. Wei et al. (2017) reported the *MEF2* gene in the Pacific white shrimp *Litopenaeus vannamei* [22], and found that *MEF2* was strongly expressed during the limb bud stage, suggesting that *MEF2* played an essential role in the early development of penaeid shrimp. Another study found that *MEF2* began to express after mesoderm proliferation and persisted in developing the musculature of the amphipod crustacean *Parhyale hawaiensis*, which is believed to activate muscle differentiation and promote muscle growth and development [23]. In banana shrimp *Fenneropenaeus merguiensis*, it was reported that the 2 kb upstream promoter region of the *FmMSTN* contained putative response elements of *MEF2* [24]. However, the expression profile and function of *MEF2* in adult shrimp are still unknown.

In this study, by analyzing the gene structure, expression profile, and the effect of RNA interference, we explored the characteristics and functions of the *MEF2* gene in *L. vannamei*, which provided an important basis for the further study of the *MEF2* gene in shrimp and crustaceans.

## 2. Results

### 2.1. Structural Analysis of the LvMEF2 Gene

A total of 17 transcripts, annotated as the *MEF2* gene, were obtained from the genome and multiple transcriptome data of *L. vannamei*. Gene location analysis showed that these sequences were all mapped on the same gene (*LvMEF2*, or *LVAN10676*), and that the 17 transcripts were named *MEF2-1* to *MEF2-17* (Appendix A). A gene structure analysis showed that most of these transcripts’ CDS (coding sequence) region was composed of seven exons, except *MEF2-12* and *MEF2-13* (Figure 1A). These findings showed that *LvMEF2* had a variety of alternative splicing forms with different types, such as alternative promoter, alternative terminator, alternative 5′ splice site, alternative 3′ splice site, exon skipping, and intron retention. Here, we focused on the alternative splicing of the CDS region. By mapping to the genome of *L. vannamei*, it was found that there were three main alternative splicing sites in the CDS region of *LvMEF2*, including a mutually exclusive exon 2 and an alternative 5′ splice site in exon 4 and exon 5. Moreover, these exons can be arranged in different combinations, and thus far, eight different CDS transcripts forms of *LvMEF2* have been found (Figure 1B).

The 17 transcripts were translated into amino acids to obtain their protein sequences. Through multiple sequence alignment, it was found that there were three main protein sequence difference regions (Appendix A) and that the corresponding positions were the site of alternative splicing. The 17 deduced proteins all had a typical MADS-MEF2 domain and a HTURP-C domain (Figure 2A), except for *MEF2-12* and *MEF2-13*. Therefore, a typical *LvMEF2* consists of 7 exons. The *MEF2-16* and *MEF2-17*, were mutually exclusive exons for splicing and were named *LvMEF2-Ι* and *LvMEF2-ΙI.*

By comparing the functional domains of the *MEF2* genes in different species, the MADS domain showed extensive sequence similarity, the HJURP-C domain was less conserved, and the C-terminal sequence had diverged considerably (Figure 2B). A multiple sequence alignment of the *MEF2* genes for different species is shown in Appendix A.

The *MEF2-16* (*LvMEF2-Ι*) and *MEF2-17* (*LvMEF2-ΙI*) were selected to analyze their physicochemical properties. *LvMEF2-Ι* encoded 428 amino acids with an estimated molecular weight of 46.67 kDa, they were mainly composed of proline (Pro, 12.1%) and serine (Ser, 12.1%); the theoretical isoelectric point (p*I*) was 8.95, the average hydrophobicity was −0.815, and the instability coefficient was 61.85. *LvMEF2-ΙI* encoded 428 amino acids with a predicted molecular weight of 46.7 kDa, they were mainly composed of proline (Pro, 12.1%) and serine (Ser, 12.6%), the theoretical isoelectric point (p*I*) was 8.82, the average hydrophobicity was −0.804, and the instability coefficient was 58.48.

### 2.2. Phylogenetic Analysis of LvMEF2 Gene

Deduced amino acid sequences of two splice variants of *LvMEF2* (LvMEF2-Ι and LvMEF2-ΙI) and 27 MEF2 protein sequences from different species (Table 1) were selected for phylogenetic tree construction. Through phylogenetic analysis, it could be found that the four paralogs of MEF2 in vertebrates had different evolutionary rates (Figure 3). Among them, the MEF2B of vertebrates belongs to an evolutionary group alone, which is in a relatively primitive clade and clustered together with nematode *Caenorhabditis elegans* and trematode *Clonorchis sinensis*. However, the remaining three paralogs of MEF2 in vertebrates were closely related and belonged to the other evolutionary group. The MEF2 of arthropods were clustered well together and constituted an independent clade, which was relatively close to the clade of MEF2A, C, and D of vertebrates, as well to as the clade of echinoderms and amphioxus.

### 2.3. LvMEF2 Gene Expression Profiles

We analyzed the expression level of *LvMEF2* in different adult shrimp tissues and found that *LvMEF2* was expressed in all 13 detected tissues. The highest expression was in the Oka organ, followed by the heart, abdominal nerve, muscle, and intestine (Figure 4A). Through transcriptome data from different tissues, we found that *LvMEF2* expressed in various splice variants and showed higher expression in the Oka, neural tissue, muscle, and intestine (Figure 4B). Among them, *MEF2-17* was expressed in the Oka organ, muscle, brain, epidermis, hemocytes, and a few other tissues, while *MEF2-16* was highly expressed in all detected tissues. Interestingly, *LvMEF2* contained several specific splice variants in different molting stages. For example, *MEF2-4* was expressed in C, D2, and P1 stages, *MEF2-5* was expressed in D3 and P1 stages, *MEF2-9* was only expressed in D1 stage, *MEF2-17* was only expressed in D3 stage, and some splice variants were barely expressed during the molting phase, such as *MEF2-6* (Figure 4C). During infection by different pathogens, many splice variants of *LvMEF2* were differentially expressed. More interestingly, whether or not infected by a pathogen, *MEF2-9* was only expressed in hemocytes, *MEF2-11* was definitely expressed in the Oka and occasionally in hemocytes, and *MEF2-17* also showed an especially high expression in the Oka but not in hemocytes. In addition, the types of *LvMEF2* splice variants were more in the Oka than in hemocytes when stimulated by pathogens (Figure 4D). At the same time, during early development, *LvMEF2* was highly expressed in limb bud stage, and the most types of splice variants were also found in the limb bud stage (Figure 4E). Then, *LvMEF2* expression was detected from the cephalothorax samples of *L. vannamei* from 20 to 80 days after post-larvae (P20-P80) by qPCR, the expression level of *LvMEF2* fluctuated, and the overall trend was gradually decreased with the growth of the shrimp (Figure 4F).

### 2.4. LvMEF2 RNA Interference

Before conducting the formal RNA interference experiment for *LvMEF2*, the optimal RNA double-stranded dose was explored using a pre-experiment. By RT-qPCR, the highest interference efficiency of 60% was found when the dosage was 2 μg/individual (Appendix A). According to the pre-experimental dose, the experimental shrimp was continuously interfered with for half a month. The results showed that a continuous injection of dsMEF2 RNA significantly down-regulated the expression level of *LvMEF2* in muscle (Figure 5A). After half a month of RNA interference, the body length and weight gain in the experimental group (dsMEF2) was significantly lower than that in the two control groups (PBS and EGFP(Enhanced Green Fluorescent Protein) groups) (Figure 5B,C). By recording the cumulative number of deaths, it could be found that the survival of the shrimps had been affected after *LvMEF2* RNA interference (Figure 5D). During the experiment, we also counted the molting number of shrimps, which showed a delayed peak during the interference (Appendix A).

In order to understand the effect of *LvMEF2* knockdown on shrimp muscle, the expression of muscle-related genes was detected. The result showed that the expressions of fast-type skeletal muscle actin (FSK) and slow-type skeletal muscle actin (SSK) genes were all significantly down-regulated after RNA interference compared with the control groups (Figure 6), which showed that *LvMEF2* RNA interference can affect muscle growth and development, both FSK and SSK.

### 2.5. Histology and Total Viable Bacteria Count after LvMEF2 Knockdown

In the RNA interference experiment, after the third injection (9 days), the shrimp of the dsMEF2 group showed a decline in vitality, had a reduced food intake, and even died. Muscle samples were sampled at the end of the interference experiment and it was found that the muscle of the dsMEF2 group was softer than the control groups. Furthermore, the abdominal muscles were sampled for sections to observe the histological changes. The result showed that the muscles of the dsMEF2 group had nuclear agglutination and an increased tissue interspace (Figure 7A,B), indicating that the muscle structure may have been impaired. The hepatopancreas were sampled for their total viable bacteria count to examine the bacterial count in the dsMEF2 and control groups. The results showed that the conditional bacterial count in the dsMEF2 group was much higher than that of the control groups after *LvMEF2* interference (Figure 7C,D).

### 2.6. Differentially Expressed Genes (DEGs) in Muscle after LvMEF2 Knockdown

After *LvMEF2* interference, the muscle of the experimental and control group was used for RNA-Seq sequencing and analysis. A total of 570 DEGs were identified, including 358 up-regulated genes and 212 down-regulated genes (Figure 8A). The up-regulated and down-regulated DEGs were mainly focused on muscle composition, immune response, protein synthesis and mitochondrial function, etc. (Table 2). Ten differential genes were chosen for expression verification, which confirmed the accuracy of the RNA-Seq results (Appendix A).

To understand the function of DEGs after *LvMEF2* interference in shrimp, all genes were mapped to terms in the GO database (Figure 8B). In the biological processes, DEGs were mainly concentrated in GO terms of “small molecule metabolic processes”, “cellular amino acid metabolic processes”, “organic acid metabolic processes”, “carboxylic acid metabolic processes”, “oxoacid metabolic processes”, “protein folding”, etc. The enriched DEGs were mainly aspartate–tRNA synthetase, glutamine–tRNA synthetase, phenylalanine–tRNA ligase, glutamate dehydrogenase, tyrosine–tRNA ligase, valine–tRNA ligase, and heat shock proteins. Thus, with the *LvMEF2* knockdown, the protein synthesis processes were affected, in which the expression of genes related to amino acid translation was reduced, and protein synthesis was inhibited.

In the cellular component aspects, DEGs were mainly enriched in GO terms of “organelle components”, “myosin complex”, “actin cytoskeleton”, “cytoskeletal components”, “mitochondrial membrane envelope”, etc. The corresponding differential genes were *Myosin N-terminal SH3-like domain*, *Myosin-4*, *Myosin heavy chain 13* (skeletal muscle), *mitochondrial import receptor*, and other genes. These results showed that the expressions of muscle composition genes were affected, and myocyte differentiation was impaired. The expression of genes, such as the mitochondrial import receptor, was down-regulated, indicating that the mitochondrial membrane in muscle was impaired and energy metabolism was inhibited after *LvMEF2* was knocked down.

In the molecular function aspects, the DEGs were mainly enriched in “pyrophosphatase activity”, “hydrolase activity”, “cation binding”, “NAD^+^ ADP-ribosyltransferase activity”, “transferase activity”, “endonuclease activity”, “aminoacyl-tRNA ligase activity” and other GO terms, and the enriched differential genes were mainly *Aspartate-tRNA ligase*, *glutamine ligase*, *Myosin heavy chain*, *Myosin-4*, *Myosin-13*, etc. The results showed that the interference of *LvMEF2* affected the binding of amino acids to tRNA and the polypeptide chain formation in muscle.

Signaling pathways were defined using the KEGG database, and the top 20 pathways are shown in Figure 8C,D. Among them, the top five pathways (enriched for up-regulated DEGs) and top three pathways (enriched for down-regulated DEGs) had significant differences after *LvMEF2* interference. In up-regulated pathways, the major differential genes enriched in the “alanine, aspartate, and glutamate metabolic” pathways were glutamyl-alanine transaminase, glutaminase, glutamate dehydrogenase, and glutamine synthetase. Pyruvate was provided by glycolysis in the muscle, and glutamate and pyruvate undergo transamination to form alanine, which was used by the organism as a carrier of ammonia from the muscle to the liver as an expression of the organism’s economy in maintaining life activities. The up-regulation of the alanine metabolic pathway indicates that metabolic activity in the muscle of *L. vannamei* was affected by *LvMEF2* interference, and that the interconversion between alanine and pyruvate was reduced.

In addition, after *LvMEF2* was knocked down, “eukaryotic ribosome formation”, “aminoacyl-tRNA biosynthesis”, and “protein processing” pathways related to the endoplasmic reticulum were down-regulated, and the main differential genes were tyrosine kinase, serine/threonine protein kinase, ribosomal proteins, and aspartate/glutamine/tyrosine/phenylalanine/valine–tRNA ligase genes. The tRNA ligase genes were incorporated into the polypeptide chain at the ribosome and covalently formed aminoacyl-tRNA with the transfer RNA, and with the aminoacyl-tRNA, bound to specific sites in the mRNA. After *LvMEF2* was disturbed, the expression of tRNA ligase genes was down-regulated, indicating that the associated amino acids were unable to bind properly to tRNA and protein synthesis was blocked. We also found the up-regulation of gene expression in related immune pathways; the corresponding genes were *immunoglobulins*, *heat shock proteins*, *NFX1-type zinc finger-containing protein 1*, *interferon-related developmental regulator*, *MFS-1*, *Ras family-related proteins*, etc., which were related to the immunity of shrimp.

## 3. Discussion

So far, there have been numerous studies on the regulation of *MEF2* genes in terms of the muscle growth and development of vertebrates and model organisms. Many other studies have also reported that *MEF2* is closely associated with neurodevelopment and the development and progression of certain tumors. However, the structure and function of the *MEF2* gene in crustaceans have been still largely unclear. In this study, based on the identification of alternative splice variants, an analysis of expression profiles, an RNA interference experiment, muscle tissue histology, and a transcriptome analysis of the *MEF2* gene in *L. vannamei*, we gained a preliminary understanding of the expression profiles and possible function of *MEF2* in shrimp.

### 3.1. LvMEF2 Has Multiple Splice Variants

Alternative RNA splicing is a way to diversify gene function. In *L. vannamei*, although only one *MEF2* gene is present in the genome, we found numerous splice variants of this gene. The alternative splice types of *LvMEF2* were mainly exon exclusivity and alternative 5’ splice site, and the different exons could be arranged in different combinations, making a boom of its alternative splicing transcripts. In vertebrates, there are multiple transcripts in *MEF2A*, *C*, and *D* genes, most of which include muscle-specific splicing with a conserved MADS-MEF2 domain and HJURP-C domain [25]. However, the MEF2Bs lack muscle-specific splice variants and several conserved domains (they only have the MADS-MEF2 domain) [26,27]. In this study, domain prediction showed that *LvMEF2* contained a MADS-MEF2 domain and a HJURP-C domain. The phylogenetic analysis showed that *LvMEF2* clustered well with the *MEF2* of crustaceans and was closely related to the vertebrate MEF2A, C, and D family, but distantly related to MEF2B. The diversity of the alternative splice variants of the *MEF2* gene enables functional diversification. Therefore, we speculated that the diversity in the *LvMEF2* alternative splice variants of *L. vannamei* corresponded to functional diversity, which deserves further study.

### 3.2. LvMEF2 Has Complex Expression Profiles

The temporal and spatial expression of *LvMEF2* was also complex and diverse. *LvMEF2* was mainly expressed in the early stage of *L. vannamei*, and its expression level gradually decreased as juvenile shrimp matured, indicating that *LvMEF2* might mainly play a role in the early muscle development of shrimp. This pattern was similar to the expression of *MEF2* gene in vertebrates. In the processes of the early muscle development stage of the Nanyang cattle [28], *MEF2A* and *MEF2D* were highly expressed, and then their expression levels gradually decreased, showing an apparent tendency to ascend first and descend later. In *L. vannamei*, the expression of *LvMEF2* could be detected in the muscle, ventral nerve, eye stalk, hepatopancreas, heart, and almost all tissue, consistent with the expression profiles of many vertebrates and invertebrates, such as zebrafish [29], crap [6] and scallop [30]. In this study, the expression profiles of the *LvMEF2* splice variants were analyzed in different tissues, different developmental stages, different molting stages, and with infection by different pathogens. The results showed that *LvMEF2* was not only ubiquitously expressed in different tissues, but that most of the splice variants were expressed. The muscle, intestine, and the Oka tissues contained the most alternative splicing, indicating that *LvMEF2* plays a crucial role in muscle growth, food digestion, and immune processes [31,32].

In comparison to *LvMEF2-Ι* and *LvMEF2-ΙI*, which were two typical alternative splicing transcripts in *L. vannamei*, *LvMEF2-Ι* was expressed in all the detected tissue and also expressed in every developmental stage, whereas *LvMEF2-ΙI* was only expressed in several special tissues and in special early developmental stages, suggesting that *LvMEF2-ΙI* may play a more specific role. Some of the splice variants (*MEF2-4/5/9/10/17*) showed specific expression at different molt stages. Interestingly, the abundant expression of *LvMEF2* transcripts was detected in the main immune tissues (Oka and hemocytes) of the shrimp after infection with different pathogens, and many of the splice variants were found to be specifically expressed in the Oka only, especially *MEF2-11* and *MEF2-17*(*LvMEF2-ΙI*), which were highly specific in the Oka. However, there were few splice variants present in hemocytes, and only *MEF2-9* was highly expressed in hemocytes. It was reported that *MEF2* played a role in the development of human muscle, heart, bone, vascular, nerve, hemocyte, and immune system cells, and influenced cell proliferation, differentiation, migration, apoptosis, and metabolism [33]. In mice, *MEF2C* was a regulator of B-cell homeostasis, and the absence of *MEF2C* in hemocytes led to a higher proportion of prophase B-cell apoptosis and necrosis [34]. The splice variants of *LvMEF2* were hypothesized to play different roles in the immune process of shrimp, which is worth more in-depth attention.

### 3.3. LvMEF2 Affects the Growth and Immunity of the Shrimp

*MEF2* genes produce different proteins by alternative splicing, and these proteins can undergo post-translational modifications and interact with diverse partner proteins, thus further performing different functions. In this study, after half a month of *LvMEF2* interference, the experimental group shrimp showed significant inhibition in their body length and weight gain, and shrimp had even died. Histological observation showed that the muscle structure of the shrimp was abnormal after *LvMEF2* knockdown, and that the expression of muscle genes was significantly down-regulated; these results indicate that the long-term interference of the *MEF2* gene could inhibit the expression of downstream muscle genes, and in turn, affect the growth of shrimp. In GO analysis, many genes, such as *ligase*, *myosin*, and *actin*, were differentially expressed. In mice, the deletion of *MEF2C* led to the disintegration of the sarcomere, suggesting that *MEF2* activity was associated with the regulation of the cytoskeletal structure of skeletal muscle [35]. Another study examined MEF2A DNA binding sites in mouse cardiac myocytes and found that candidate target genes were mainly enriched in the functional pathways associated with cardiac and muscle development and cytoskeletal organization [36]. At the same time, we found that the genes associated with mitochondria were significantly down-regulated after *LvMEF2* interference. The *MEF2* gene of yeast *Saccharomyces cerevisiae* has been reported to be a mitochondrial protein translation factor, and evidence suggests that it participates in ribosome recycling and plays an important role in maintaining mitochondrial health, which may also be shared by other mitochondrial protein synthesis factors [37]. Therefore, when *MEF2* is knocked down, organelles in muscle (especially mitochondria) may be damaged.

KEGG analysis showed that the DGEs were significantly enriched in metabolism pathways, especially protein synthesis-related pathways, such as “alanine, aspartate and glutamate metabolism”, “ribosome biogenesis in eukaryotes”, “aminoacyl-tRNA biosynthesis”, and “protein processing in the endoplasmic reticulum”. The changes indicated that the protein synthesis of *L. vannamei* was drastically affected after *LvMEF2* interference. It has been reported that *MEF2* is a regulator of muscle fiber homeostasis when the *MEF2* function is weakened, resulting in the expansion of glutamine, which leads to skeletal muscle atrophy [38]. In *Drosophila*, the mutation of a single *MEF2* gene results in muscle cells that do not differentiate [39]. Previous studies have also shown that *MEF2* plays a crucial role in the early stage of myoblast fusion, and myoblasts lacking the *MEF2* gene cannot fuse into myotubes normally [40]. In addition, *MEF2* have roles in regulating apoptosis, in the expression of *MEF2* in endothelial cells and in the development of neurons inhibiting apoptosis downstream of MAPK signaling [41,42].

In *L. vannamei*, in addition to having a higher expression in muscle, *LvMEF2* expression was highest in the Oka organ. A transcriptome analysis of muscle samples for 48 h of *LvMEF2* interference showed that protein synthesis and mitochondria were damaged, and that the expression of related genes in the immune pathway was up-regulated after *LvMEF2* knockdown. These results suggest that *MEF2* played an important role not only in muscle formation and growth, but also in the immunity of shrimp. It was reported that *MEF2* was phosphorylated at a conserved site in healthy *Drosophila*, which promoted the expression of lipogenic enzymes and glycogen synthase kinase. However, when infected by pathogens, this phosphorylation would be lost, and the activity of *MEF2* changes and promotes the production of antimicrobial peptides [43]. Therefore, *MEF2* was considered to be a key transcriptional switch between metabolism and immunity in adult *Drosophila*, and similar processes may exist in shrimp.

## 4. Materials and Methods

### 4.1. Experimental Animals

The healthy shrimp *L. vannamei* used in this experiment were cultured in the shrimp breeding laboratory of the aquarium building, Institute of Oceanology, Chinese Academy of Sciences. The shrimp used in this experiment had a body length of 7.5 ± 0.5 cm and a body weight of 4.5 ± 0.5 g. Before the experiment, the shrimp were cultured in the breeding tank for a week, and the temperature of the aerated seawater was maintained at 25 ± 1 °C, with a salinity of 30‰ and pH of 7.5 ± 0.1. During the experiment, seawater was changed once a day, and shrimp were fed regularly and quantitatively with commercial food pellets thrice a day. All experimental shrimp were handled and treated according to the guidelines approved by the Animal Ethics Committee [2020(37)] at the Institute of Oceanology, Chinese Academy of Sciences (Qingdao, China). No rare or endangered animals were used in this study.

### 4.2. Identification and Analysis of MEF2 Gene in L. vannamei

To identify the *MEF2* gene in *L. vannamei*, all annotated *MEF2* sequences were extracted from the genome (http://www.shrimpbase.net/lva.dowload.html, accessed on 15 July 2021) and transcriptome data obtained previously in our laboratory from different tissues, different early development stages, different molting stages and the three pathogens (*Staphylococcus aureus*, *Vibrio parahaemolyticus* and white spot syndrome virus (WSSV)) [22,44,45]. The ORF Finder (https://www.ncbi.nlm.nih.gov/orffinder/, accessed on 19 July 2021) and ExPASy translate tool (https://web.expasy.org/translate/, accessed on 19 July 2021) were used to obtain the corresponding amino acid sequences and the corresponding ranges of CDS regions for these transcripts. GSDS2.0 (http://gsds.gao-lab.org/, accessed on 25 July 2021) was used to analyze and map the gene structure of the transcripts. The domains were analyzed for these transcripts through the online website InterProscan (https://www.ebi.ac.uk/interpro/about/InterProscan/, accessed on 10 November 2022). Various forms of alternative splicing (Appendix A) were identified by mapping to the genome sequences. The two longest nucleotide sequences in the typical splice variants were selected for additional structural and phylogenetic analyses.

### 4.3. Construction of Phylogenetic Tree

To understand the phylogeny of the *MEF2* gene in *L. vannamei* and other species, a total of 26 MEF2 amino acid sequences (Appendix A) from representative organisms in different taxa were downloaded from NCBI (https://www.ncbi.nlm.nih.gov/, accessed on 15 October 2022). The sequence IDs used in the phylogenetic analysis are shown in Table 2, with 20 sequences from invertebrates and 9 sequences from vertebrates. The MEGA 11 platform was used for the construction of phylogenetic trees, and the MEF2 sequences of different species were aligned using the Cluster W algorithm under the default model. The phylogenetic tree was constructed based on the neighbor-joining (NJ) method using the default values for all parameters. Finally, the phylogenetic tree was visualized and ornamented through the I-TOL tree online website (https://itol.embl.de/, accessed on 22 October 2022 ).

### 4.4. RNA Extraction and cDNA Synthesis

RNA was isolated using RNAiso Plus Reagent (Takara Bio Inc., Kyoto, Japan) according to the manufacturer’s instructions. The quality and concentration of RNA were detected using 1% agarose gel electrophoresis and Nano Drop 2000 (Thermo Fisher Science, Waltham, MA, USA), respectively. Then, first-strand cDNA was synthesized by reverse transcription-polymerase chain reaction (RT-PCR) using 1 μg of RNA and the Prime Scrip First Stand cDNA Synthesis kit (Takara, Kyoto, Japan). The reaction was carried out in two steps. In the first step, a gDNA eraser was used to eliminate DNA from the genome, and the reaction was carried out at 42 °C for 5 min. In the second step, cDNA was synthesized using reverse transcription, and the reaction was performed at 37 °C for 1 h and 85 °C for 5 s. The synthesized cDNA was stored at −20 °C.

### 4.5. Gene Expression Pattern Analysis

In order to detect the expression of the *LvMEF2* in *L. vannamei*, RNA was extracted from different adult tissues, and the cephalothorax of seven different developmental stages after post-larvae; then, the reverse transcription and quantitative fluorescence PCR (qPCR) were performed. The primers were shown in Appendix A. The qPCR reaction conditions were as follows: firstly, pre-denaturation for 30 s at 95 °C, and then at 95 °C for 5 s, followed by the annealing temperature (according to the primer annealing temperature) for 35 s; this was repeated for a total of 40 cycles. The 2^−ΔΔCt^ method [46] was used for relative quantitative analysis. The data were analyzed by one-way analysis of variance using GraphPad Prism 9.3.1.471 software. Different letters were used to represent significance levels. The expression heat maps of the splice variants of *LvMEF2* in different tissues, different developmental periods, different molting periods, and infection with different pathogens were drawn by TBtools V1.098 software based on the transcriptome data of our previous studies [22,44,45].

### 4.6. Double-Stranded RNA Synthesis and Interference Experiments of LvMEF2

Double-stranded RNA (dsRNA) with a length of 500 to 600 bp was designed based on the conserved domains of *LvMEF2* and *EGFP* (enhanced green fluorescent protein, as control). Primers were designed to amplify them (Appendix A). Double-stranded RNAs were synthesized in vitro using the TranscriptAid T7 High Yield Transcription Kit (Thermo Scientific, Vilnius, Lithuania) and the primers containing the T7 promoter; the annealing temperatures are shown in Appendix A. The quantitative real-time PCR was run on Eppendorf Mastercycler ep realplex (Eppendorf, Hamburg, Germany). The THUNDERBIRDTM SYBR^®^ qPCR Mix Without ROX kit (Toyobo, Osaka, Japan) was used for qRT-PCR with the primers named MEF2-YG-F and MEF2-YG-R. The 18S was used as an internal reference. The 10 μL system contained 5 μL of PreMix, 1 μL of template, 1 μL of upstream and downstream primers, and RNA-free water supplemented to 10 μL. The PCR procedure was as follows: firstly, pre-denaturation at 94 °C for 5 min, then denaturation at 94 °C for 30 s, annealing at 60 °C for 30 s, extension at 72 °C for 40 s by 40 cycles, and at last, an extension of 10 min.

A total of 84 shrimp were used in the pre-experiment with the body weight of 4.56 ± 0.5 g, respectively. The purified and qualified dsRNA was diluted with 1 × PBS. The experimental group was named dsMEF2 (injected with *LvMEF2* dsRNA), the dsEGFP group (injected with *EGFP* dsRNA) and PBS (solvent) group were used as controls. Each group was set up with three biological replicates (four individuals/replicate) with three gradients of 0.5 μg, 1 μg, and 2 μg. Then, dsRNA 10 µL of solvent was injected into the last abdominal segment of each shrimp separately. At 48 h post-dsRNA injection, the RNA was extracted from muscle, and the dsRNA interference efficiency was detected. The dosage of 2 µg of dsRNA per individual was selected for further RNAi experiments. In the formal experiment, a total of 216 individuals in the pre-molt stage (D1–D2) were randomly divided into three groups (the experimental group, the dsEGFP and PBS control groups). The body length and body weight of every shrimp were measured before injection with the optimal dose. The formal interference experiment lasted for half a month; the same injection was repeated every 4 days to keep the gene interference efficiency. After the experiment, the same method was used to measure the body weight and body length again. Then, abdominal muscles were sampled and frozen in liquid nitrogen and then stored at −80 °C, RNA was extracted, and the expression of *LvMEF2* and muscle-related genes was detected by quantitative fluorescence PCR (qPCR). The data were analyzed by the *t*-test using GraphPad Prism software. *p* < 0.05 was marked as a single asterisk.

### 4.7. Tissue Section and Total Viable Bacteria Count after LvMEF2 Gene Knowdown

After half a month of interference, the muscle tissue in the second abdominal segment of *L. vannamei* was sampled for section preparation. Briefly, the tissue was fixed with 4% paraformaldehyde and dehydrated with different concentrations of ethanol. Then, the tissue was treated with xylene and embedded in paraffin (Sigma, Roedermark, Germany). The embedded tissues were sliced with a microtome, and the thickness of the slices was about 7 μm. Then, the sections were stained with hematoxylin-eosin (HE) and examined under a microscope.

In order to detect the total viable bacteria count in shrimp hepatopancreas after interference, the hepatopancreas of 6 shrimp were sampled from the experimental and control groups, respectively. The collected hepatopancreas samples were weighted, crushed, and blended in sterile PBS separately. Then, 100 μL of the suspension with a 5-fold dilution was seeded onto TCBS agar. After incubation at 28 °C for 18 h, the number of total bacterial colonies on the plate was counted.

### 4.8. Transcriptome Sequencing Analysis of Muscle Samples after LvMEF2 Interference

Muscle samples from *L. vannamei* after *LvMEF2* interference were sampled for transcriptome sequencing. Total RNA was separately extracted using the Trizol reagent kit (Invitrogen, Waltham, MA, USA) according to the manufacturer’s instructions. The RNA completeness and concentration were checked using 1% RNase-free agarose gel electrophoresis and Agilent 2100 Bioanalyzer (Agilent Technologies, Santa Clara, CA, USA). Subsequently, library construction and quality inspection were conducted. mRNA was enriched by Oligo (dT) beads and then fragmented and reverse-transcribed into first-strand cDNA with random primers. After the second-strand cDNA was synthesized, the cDNA fragments were purified using a QiaQuick PCR extraction kit (Qiagen, Hilden, Germany). The synthesized cDNA fragments were repaired, an adapter was added and specific length fragments (generally 250–300 bp) were recovered to complete the construction of the cDNA library. After the library was constructed, qRT-PCR was used to accurately quantify the effective concentration of the library. Finally, the paired-end cDNA library was sequenced using Illumina NavoSeq 6000 platform (Illumina, San Diego, CA, USA) in Novogene Corporation Inc (Beijing, China).

To obtain high-quality clean reads, adapter sequences and low-quality reads were filtered by fastp, and the ribosome RNA (rRNA) mapped reads were further removed by Bowtie2 (version 2.2.8) [47]. All raw reads were deposited on the NCBI Sequence Reading Archive (SRA) website (SRR23060691-SRR23060696). The remaining paired-end clean reads of each sample were separately mapped to the shrimp genome [45] using HISAT2 (version 2.4) [48] with default parameters. Then, the mapped reads of all samples were assembled using StringTie (version 1.3.1) [49] with on a reference-based approach. To quantify the expression abundance, a FPKM (fragment per kilobase of transcript per million mapped reads) value for each transcription region was calculated using StringTie software [50].

To estimate the change in the transcript levels after *LvMEF2* interference, a differential expression analysis of the gene expression between the experimental and control groups was performed using DESeq2 software (1.20.0) with padj ≤0.05 and|log_2_(foldchange)| ≥ 1 as a significant difference. In order to illustrate the DEG cell composition, molecular function, and biological processes, the Novogene online tools (https://magic.novogene.com/, accessed on 7 November 2022) continued the GO and KEGG enrichment analysis. The expression of DEGs was visualized by TBtools (V1.098).

In order to verify the reliability and accuracy of the transcriptome sequencing and analysis, muscle tissues from the dsMEF2 and dsEGFP groups were sampled, and four biological replicates were set up in each group. The total RNA from the samples was extracted and reverse transcribed into cDNA, as mentioned above. Ten DEGs were selected to detect the expression by RT-qPCR.

## 5. Conclusions

In this study, we identified a *MEF2* gene in *L. vannamei* and found that it has multiple splice variants. By analyzing the expression profiles of *LvMEF2* in different developmental stages, different adult tissues, different molting stages, after infection with different pathogens, as well as after RNA interference, we found that *LvMEF2* had a significant effect on the growth and development of *L. vannamei*, not only in body length and weight, but also in survival rate. Further transcriptome analysis showed that *LvMEF2* could significantly affect metabolism, growth, and immunity in the shrimp. This study provides an important basis for further investigation into the role of *MEF2* gene in the growth, development, and immune processes of shrimp and crustaceans.

## Figures and Tables

**Figure 1 ijms-24-05832-f001:**
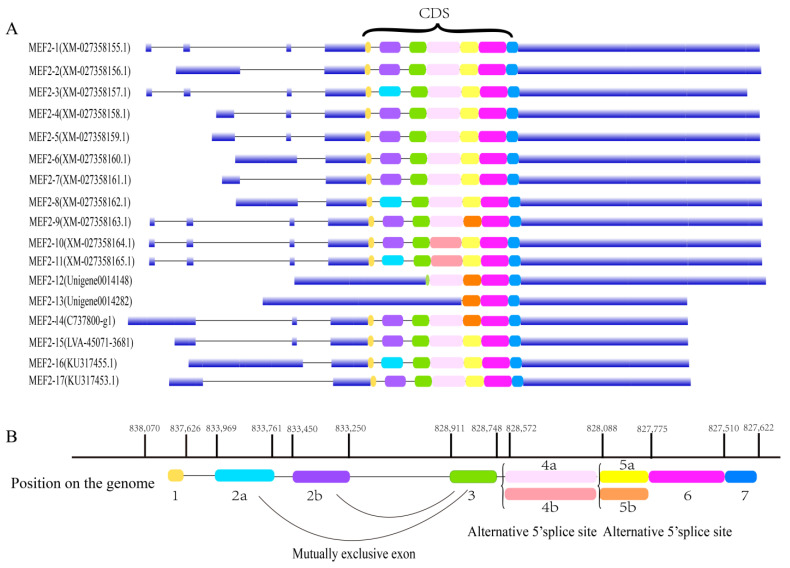
Alternative splicing of the *MEF2* gene in *L. vannamei*. (**A**) The gene structures of 17 splice variants of *LvMEF2*. Different colors in the CDS region represent different exons, and the blue blocks which outside the CDS region are the 5′UTR and 3′UTR of the genes, respectively. (**B**) Different CDS combinations of transcripts of *LvMEF2*. Two forms of alternative splice variants, a mutually exclusive exon and an alternative 5′ splice site, exist in the CDS region of *LvMEF2.* Mutually exclusive exons are two exons that cannot be present in one transcript at the same time, as 2a and 2b. Alternative
5′splice site is the splice site at the 5′end of one exon was different, resulting in an extension of the 5′end of the exon in one of the transcripts, as 4a and 4b, and 5a and 5b. There are two forms in exon 4 in which 4a is missing 18 bp base at the 5′ end of exon 4 compared with 4b. There are two forms in exon 5 in which 5a is missing 3 bp bases at the 5′ end of exon 5 compared with 5b. The numbers 1, 3, 6, and 7 indicate exons 1, 3, 6, and 7 of *LvMEF2*, respectively.

**Figure 2 ijms-24-05832-f002:**
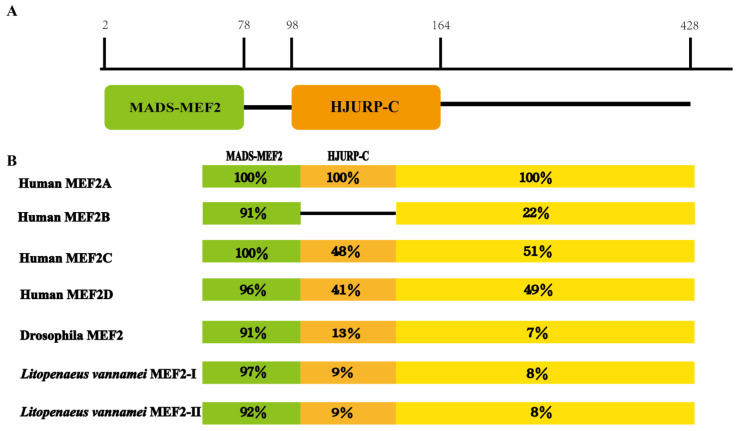
Deduced *LvMEF2* domain composition and comparison with *Drosophila* and human. (**A**) Functional domain composition of *MEF2* in *L. vannamei*. The black horizontal lines refer to the positions of amino acids that are not part of the functional domain. (**B**) Predicted *MEF2* proteins of *L. vannamei* were compared with *Drosophila* and human. Numbers represent the similarity of the domains; human MEF2A is used as a molecular model in the (**B**), and the black horizontal line shows that MEF2B has no HTURP-C domain. The green blocks refer to MADS-MEF2 domain, the orange blocks refer to HTURP-C domain, the yellow blocks refer to the C-terminus.

**Figure 3 ijms-24-05832-f003:**
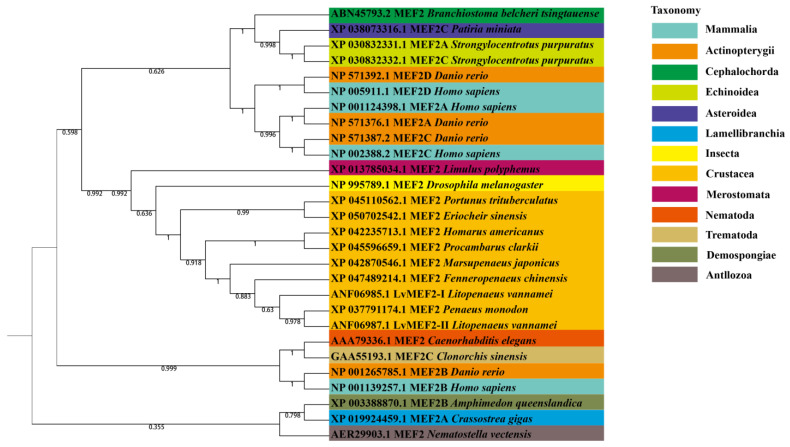
Phylogenetic analysis of *LvMEF2* and MEF2s of other species. The bootstrap values are given at each branch node. Different colors represent different classes.

**Figure 4 ijms-24-05832-f004:**
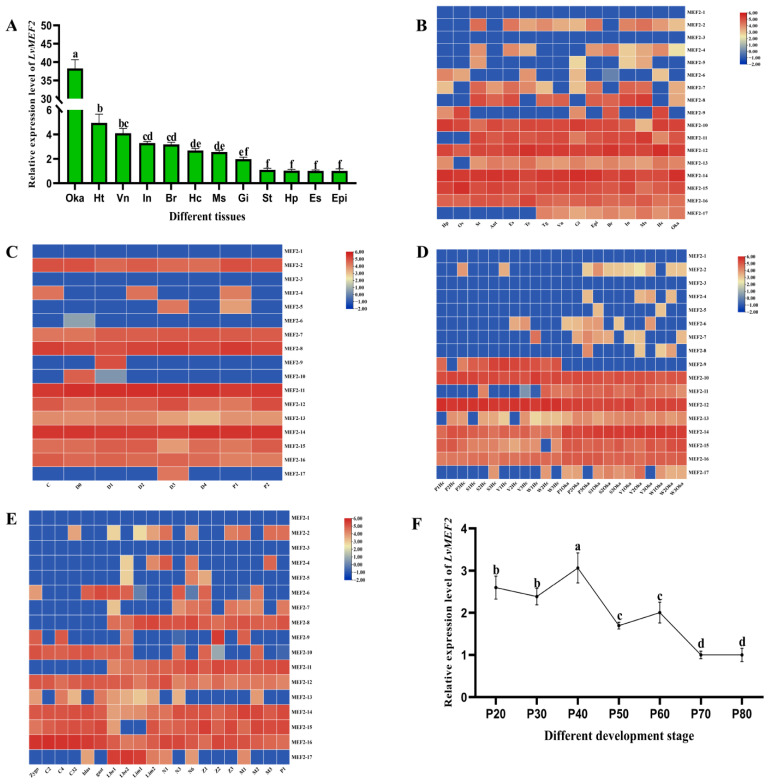
The expression profiles of *LvMEF2*. (**A**) Expression of total *LvMEF2* in different adult shrimp tissues, heart (Ht), epidermis (Epi), muscle (Ms), lymphoid organ (Oka), stomach (St), gills (Gi), eye stalk (Es), brain (Br), hemocytes (Hc), hepatopancreas (Hp), intestine (In), and abdominal nerve (Vn). These results were based on three independent biological replications and shown as mean values ± SD. Significant differences in the gene expression levels between the three treatments are shown as a, b, bc, cd, de, ef, and f. (**B**) The expressions of *LvMEF2* splice variants in different tissues. heart (Ht), epidermis (Epi), muscle (Ms), lymphoid organ (Oka), stomach (St), gills (Gi), eye stalk (Es), brain (Br), hemocytes (Hc), hepatopancreas (Hp), intestine (In), and abdominal nerve (Vn), ovary(Ov), antenna(Ant), testis(Te), thoracic ganglion(Tg). (**C**) The expressions of *LvMEF2* splice variants at different molting stages, inter molting (C), pre-molting (D0, D1, D2, D3, and D4), and post-molting (P1 and P2). (**D**) The expressions of *LvMEF2* splice variants in Oka, hepatopancreas, and hemocytes after infection by different pathogens, i.e., *Vibrio parahaemolyticus* (V), *Staphylococcus aureus* (S), and WSSV (W); sterile phosphate-buffered saline (PBS) (P) was set as the control group. Three biological replicates were taken after each pathogen infection. (**E**) The expressions of *LvMEF2* splice variants at different early developmental stages. Zygote (Zygo), 2 cells (C2), 4 cells (C4), 32 cells (C32), blastocyst (blas), gastrula (gast), limb bud embryo I (Lbe1), limb bud embryo II (Lbe2), larva in membrane Ι I (Lim1), larva in membrane Ⅱ (Lim2), nauplius I (N1), nauplius III (N3), nauplius VI (N6), zoea I (Z1), zoea II (Z2), zoea III (Z3), mysis I (M1), mysis II (M2), mysis III (M3), and post-molt stage 1 (P1). (**F**) Expression of total *LvMEF2* during the period from juvenile to adult. Cephalothoraxes of *L. vannamei* were sampled and detected after the development of post-larvae 20 to 80 days. These results were based on three independent biological replications and shown as mean values ± SD. Significant differences in the gene expression levels between three treatments are shown as a, b, c, d.

**Figure 5 ijms-24-05832-f005:**
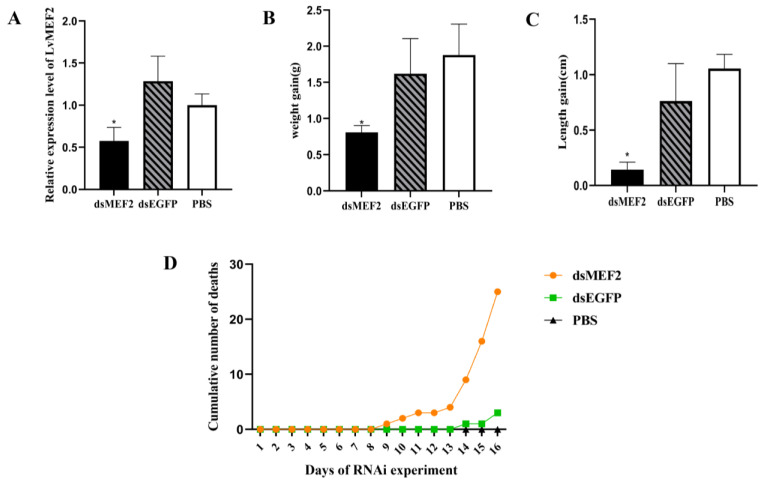
The effects of *LvMEF2* RNA interference on *L. vannamei*. (**A**) Relative expression level of *LvMEF2* in muscle after half a month of interference. dsEGFP(Enhanced Green Fluorescent Protein) and PBS were used as controls. The expression of target genes was detected by qRT-PCR, and 18S rRNA gene was simultaneously amplified as the internal reference. (**B**) Changes in shrimp body weight after half a month of RNA interference. (**C**) Changes in body length after half a month of RNA interference. These results were based on three independent biological replications and shown as mean values ± SD. Significant differences in the gene expression levels between three treatments are shown as * *p* < 0.05. (**D**) Cumulative number of deaths during the *LvMEF2* gene interference.

**Figure 6 ijms-24-05832-f006:**
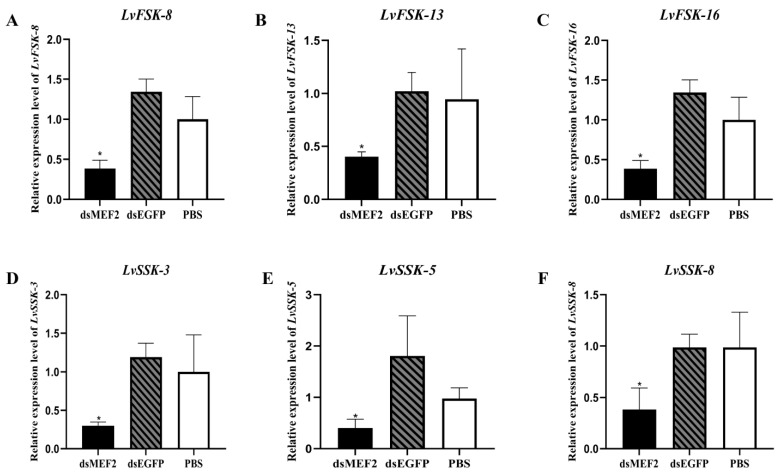
The expression of fast- and slow-type muscle genes after *LvMEF2* RNA interference. (**A**–**C**) are fast-type skeletal muscle actin (FSK), *LvFSK-8*, *LvFSK-13*, and *LvFSK-16*. (**D**–**F**) was slow-type skeletal muscle actin (SSK), *LvSSK-3*, *LvSSK-5*, and *LvSSK-8*. Significant differences in the gene expression levels between three treatments are shown as * *p* < 0.05.

**Figure 7 ijms-24-05832-f007:**
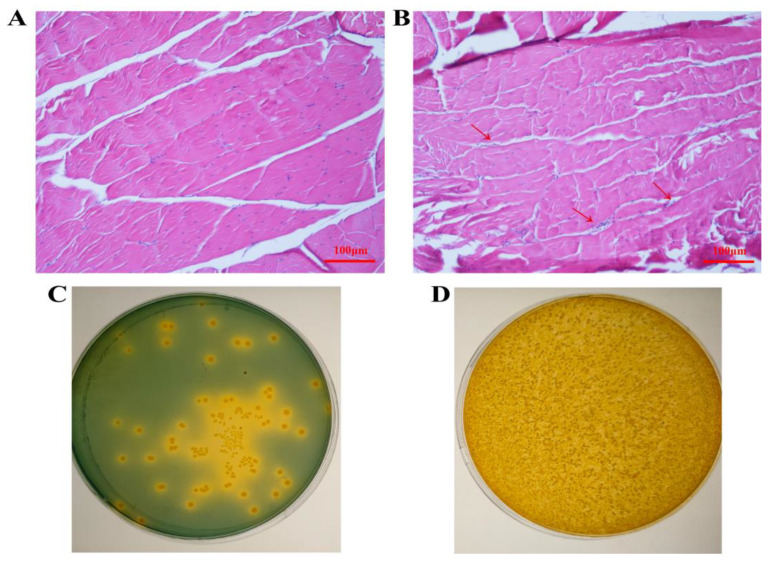
Histological sections of muscles and total viable bacteria count after *LvMEF2* interference. (**A**,**C**) control group(PBS); (**B**,**D**) dsMEF2 group. The arrows in panel (**B**) refers to the region of nuclear agglutination.

**Figure 8 ijms-24-05832-f008:**
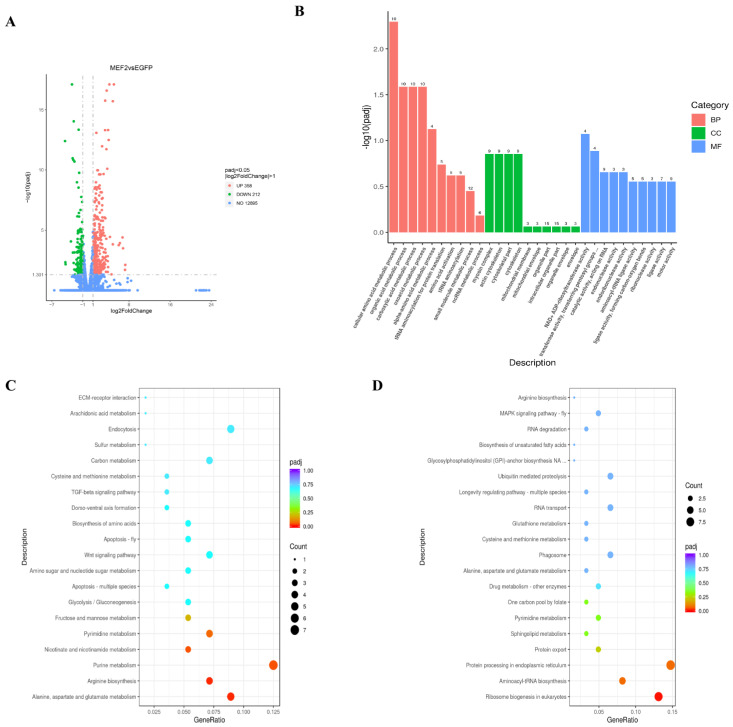
Differentially expressed genes after *LvMEF2* knockdown and their functional enrichment analysis. (**A**) Volcano plot displays DEGs between the dsMEF2 group and the EGFP control group. Blue dots indicate non-significant differential expressions, red dots indicate significantly up-regulated expressions, and green dots indicate significantly down-regulated expressions. (**B**) GO term distribution for the DEGs. The horizontal axis is the GO term, and the vertical axis is the significance level of GO term enrichment, which is represented by −log10(padj). Padj is the *p*-value after multiple hypothesis testing corrections; the higher the value, the more significant it is. Different colors represent the three GO subcategories: BP (biological process), CC (cellular component), and MF (molecular function). (**C**) The top 20 KEGG pathways enriched in muscle for up-regulated DEGs. The abscissa is the ratio of the number of differential genes annotated to the KEGG pathway to the total number of differential genes, and the ordinate is the KEGG pathway. (**D**) The top 20 KEGG pathways enriched in muscle for down-regulated DEGs.

**Table 1 ijms-24-05832-t001:** *MEF2* protein sequences for phylogenetic tree analysis.

Phylum	Class	Species	Accession Number	Name
Chordata	Mammalia	*Homo sapiens*	NP_001124398.1	*MEF2A*
			NP_001139257.1	*MEF2B*
			NP_002388.2	*MEF2C*
			NP_005911.1	*MEF2D*
	Actinopterygii	*Danio rerio*	NP_571376.1	*MEF2A*
			NP_001265785.1	*MEF2B*
			NP_571387.2	*MEF2C*
			NP_571392.1	*MEF2D*
	Cephalochorda	*Branchiostoma*	ABN45793.2	*MEF2*
Echinodermata	Echinoidea	*Strongylocentrotus purpuratus*	XP_030832331.1	*MEF2A*
			XP_030832332.1	*MEF2C*
	Asteroidea	*Patiria miniata*	XP_038073316.1	*MEF2*
Mollusca	Lamellibranchia	*Crassostrea gigas*	XP_019924459.1	*MEF2A*
Arthropod	Insecta	*Drosophila melanogaster*	NP_995789.1	*MEF2*
	Crustacea	*Litopenaeus vannamei*	ANF06985.1	*LvMEF2-Ι*
			ANF06987.1	*LvMEF2-ΙI*
		*Fenneropenaeus chinensis*	XP_047489214.1	*MEF2*
		*Marsupenaeus japonicus*	XP_042870546.1	*MEF2*
		*Penaeus monodon*	XP_037791174.1	*MEF2*
		*Procambarus clarkii*	XP_045596659.1	*MEF2*
		*Homarus americanus*	XP_042235713.1	*MEF2*
		*Eriocheir sinensis*	XP_050702542.1	*MEF2*
		*Portunus trituberculatus*	XP_045110562.1	*MEF2*
	Merostomata	*Limulus polyphemus*	XP_013785034.1	*MEF2*
Nematoda	Nematoda	*Caenorhabditis elegans*	AAA79336.1	*MEF2*
Platyhelminthes	Trematoda	*Clonorchis sinensis*	GAA55193.1	*MEF2*
Coelenterata	Antllozoa	*Nematostella vectensis*	AER29903.1	*MEF2*
Porifera	Demospongiae	*Amphimedon queenslandica*	XP_003388870.1	*MEF2*

**Table 2 ijms-24-05832-t002:** Differentially expressed genes enriched in muscle after *LvMEF2* interference.

	Gene ID	Gene Description	log_2_Fold Change
Muscle-related	novel.2155	Myosin N-terminal SH3-like domain	−4.509773837
	LVAN04176	Myosin-4	−2.978206791
	novel.4121	Myosin heavy chain 13, skeletal muscle	−1.772697548
	LVAN16923	ACT2_BOMMO Actin, muscle-type	−4.386672998
	LVAN16916	ACT3_DROME Actin-57	−2.098122282
	LVAN16917	ACT_MANSE Actin, muscle	−2.036265474
Immunity and cellular-stress-related	LVAN23842	Heat shock protein 22	−2.237662541
	LVAN23843	Heat shock protein 20	−2.036169236
	LVAN17308	Heat shock protein 70	−1.753724055
	LVAN17629	Heat shock protein 60	−1.44127178
	LVAN23851	Heat shock protein 20	−1.939231582
	novel.751	Heat shock protein 10, mitochondrial	−1.216804722
	LVAN09991	Heat shock protein 90	−1.75361939
	LVAN17366	Heat shock protein 70	−1.2135562
	novel.2459	Heat shock protein 70	−1.068649747
	novel.386	Heat shock protein 70	−1.019314088
	LVAN24387	Heat shock protein 27	−1.40954243
	LVAN23841	Heat shock protein 20/alpha-crystallin family	−1.853351031
	LVAN14669	Heat shock protein 20/alpha-crystallin family	−1.839008177
	LVAN16387	Immunoglobulin I-set domain	1.62053716
	LVAN04048	Immunoglobulin I-set domain	1.716095306
	LVAN04045	Immunoglobulin I-set domain	3.467470081
	LVAN10981	Immunoglobulin I-set domain	1.13315602
	LVAN19532	Immunoglobulin I-set domain	1.175834599
	LVAN14378	Immunoglobulin I-set domain	1.8356356
	LVAN12953	Ras-like GTP-binding protein Rho1	1.946771427
	LVAN25515	Ras-related protein Rab-2A	1.317082979
	novel.5097	NFX1-type zinc finger-containing protein 1	2.242125717
	LVAN22842	NFX1-type zinc finger-containing protein 1	1.523548395
	LVAN00304	NFX1-type zinc finger-containing protein 1	3.012657901
	novel.1678	Interferon-related developmental regulator 1	−4.437000125
	novel.1677	Interferon-related developmental regulator 2	−1.275543979
	LVAN11504	MFS-1	−2.807489076
Protein synthesis-related	LVAN04287	Glutamate–cysteine ligase catalytic subunit	−1.984075309
	LVAN14777	Methionine synthase	−1.947132225
	novel.4193	Aspartate–tRNA ligase	−1.771238279
	LVAN22403	Eukaryotic translation initiation factor 2 subunit 1	−1.674467809
	novel.1790	Tubulin beta-1 chain	−1.648057083
	LVAN06976	Glutaminyl–tRNA synthetase	−1.506118061
	novel.2347	E3 ubiquitin–protein ligase	−1.489605619
	novel.1785	Glutamic–pyruvic transaminase	−1.444998457
	LVAN22801	Serine/threonine–protein kinase	−1.278572371
	LVAN07472	Tyrosine–tRNA ligase	−1.25618501
	novel.3677	Serine/threonine–protein kinase	−1.193831772
	LVAN13244	Phenylalanine–tRNA ligase beta subunit	−1.114841361
	novel.2206	Valine–tRNA ligase	−1.072706294
	LVAN15077	Casein kinase II subunit	−1.026426438
Mitochondrial-related	LVAN00040	Mitochondrial import inner membrane translocase subunit	−2.067899986
	LVAN06526	Mitochondrial glycine transporter	−1.79981545
	novel.716	Mitochondrial pyruvate carrier 1	−1.480260336
	novel.2831	Mitochondrial import receptor subunit	−1.018535966
	novel.2485	Mitochondrial-processing peptidase subunit alpha	−1.017153841

## Data Availability

The datasets presented in this study can be found in online repositories. The names of the repository/repositories and accession number(s) can be found in the article/Appendix A.

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
