# Peer review of "Gene Structure, Expression and Function Analysis of MEF2 in the Pacific White Shrimp Litopenaeus vannamei"

_ijms, 2023, doi:10.3390/ijms24065832_

Round 1

Reviewer 1 Report

This is an important study. Xia et al. have characterized the gene structure and expression profiles of Myocyte Enhancer Factor 2 (MEF2) gene of the Pacific white shrimp Litopenaeus vannamei using genomic and transcriptomic data. MEF2 gene has an influence on the development and growth of metazoan in general. The manuscript is well-written, and the objectives and the results are clear. I believe this study will add a great contribution to the scientific community. Before the publication of this manuscript, I would suggest adding to the abstract a background text, as the authors didn't provide sufficient information about the targeted gene or the species of the study.

Author Response

This is an important study. Xia et al. have characterized the gene structure and expression profiles of Myocyte Enhancer Factor 2 (MEF2) gene of the Pacific white shrimp Litopenaeus vannamei using genomic and transcriptomic data. MEF2 gene has an influence on the development and growth of metazoan in general. The manuscript is well-written, and the objectives and the results are clear. I believe this study will add a great contribution to the scientific community. Before the publication of this manuscript, I would suggest adding to the abstract a background text, as the authors didn't provide sufficient information about the targeted gene or the species of the study.

Response: Thank you for your comment and valuable suggestions. We have added a background text about the MEF2 gene and the species to the abstract as “The Pacific white shrimp Litopenaeus vannamei is the most economically important crustacean in the world. The growth and development of shrimp muscle has always been the focus of attention. Myocyte Enhancer Factor 2 (MEF2), a member of MADS transcription factor, has an essential influence on the growth and development programs including myogenesis.”

Reviewer 2 Report

In this paper the Authors investigated the role of the Myocyte Enhancer Factor (MEF2) gene in growth and development of the Pacific white shrimp L. vannamei. Specifically, they analysed its expression in different tissues and in different developmental stages, then they examined its silencing by RNA interference. They found that MEF2 is widely expressed and it shows many splice variants, some of which seem to be tissue- or developmental stage- specificic. Its RNA interference impacts on growth, both in length and in weight, and on some immune-related pathways. These results are interesting and may also represent important information for L. vannamei breeding.

There are some observations:

Fig. 2: in B I suppose the “model” molecule they refer to when describing the numbers is human MEF2A, but the text is not clear;

Fig. 4: in A there are no testes and ovary tissues in the diagram, but they are cited in the legend. I would rather list the abbreviations for the tissues independently of the panels description;

Fig. 5: in the legend, the phrases on biological replicates and significance refer to the panels A, B and C, they should be moved after the description of panel C. In the figure there is no significance below 0.01, so it is unnecessary to have it in the legend (the same is in section Materials and Methods, row 499). Moreover, panel D is unclear to me, I don’t understand what I am looking at.

Fig. 8: in the legend, please explicitate the GO subcategories

Rows 377-394 in many cases Lv is repeated twice in the gene name

A careful English editing is necessary, both in the main text and in the figure legends

Author Response

In this paper the Authors investigated the role of the Myocyte Enhancer Factor (MEF2) gene in growth and development of the Pacific white shrimp L. vannamei. Specifically, they analysed its expression in different tissues and in different developmental stages, then they examined its silencing by RNA interference. They found that MEF2 is widely expressed and it shows many splice variants, some of which seem to be tissue- or developmental stage- specificic. Its RNA interference impacts on growth, both in length and in weight, and on some immune-related pathways. These results are interesting and may also represent important information for L. vannamei breeding.

Response: Thank you for your comment concerning our manuscript. I accept your valuable suggestions and make a one-to-one response as follows.

There are some observations:

Fig. 2: in B I suppose the “model” molecule they refer to when describing the numbers is human MEF2A, but the text is not clear;

Response: Thank you for your suggestion. We have added a note in Figure 2, and the HTURP-C domain is not present in human MEF2B, we have also modified the structure of the figure from a grey box to a black line to indicate the absence of the HTURP-C domain in human MEF2B without changing the original content of the manuscript.

Fig. 4: in A there are no testes and ovary tissues in the diagram, but they are cited in the legend. I would rather list the abbreviations for the tissues independently of the panels description;

Response: We are very sorry for the negligent error here. The testes and ovary tissues have been removed from the legend in figure 4 in the revised version.

Fig. 5: in the legend, the phrases on biological replicates and significance refer to the panels A, B and C, they should be moved after the description of panel C. In the figure there is no significance below 0.01, so it is unnecessary to have it in the legend (the same is in section Materials and Methods, row 499). Moreover, panel D is unclear to me, I don’t understand what I am looking at.

Response: Thank you for your suggestion, the legend of Figure 5 has been revised. In addition, we use the line graph of cumulative death to replace that of molting, which can more clearly show the effect of MEF2 RNA interference on the experimental shrimp.

Fig. 8: in the legend, please explicitate the GO subcategories

Rows 377-394 in many cases Lv is repeated twice in the gene name.

Response: Thank you for your suggestion, we added the annotation of Go subclass in the legend of Fig. 8. as “Different colors represent the three GO subcategories BP (biological process), CC (cellular component), and MF(molecular function).” The excess “Lv” in gene names has been removed.

A careful English editing is necessary, both in the main text and in the figure legends

Response: We thanks for your valuable suggestions and comments, which provided great help for our manuscript. We have checked the language of the revised manuscript carefully.

Reviewer 3 Report

Reviewers' Comments to Authors:

The manuscript entitled “Gene Structure, Expression and Function Analysis of MEF2 in the Pacific White Shrimp Litopenaeus vannamei by Xia et al. demonstrates experimental attempts to molecularly characterize and identify gene structure, expression, and functional analyses of the myocyte enhancer factor 2 (MEF2) of Pacific white shrimp in various conditions. The results indicate that MEF2 was intensely involved in muscle formation and immune responses of shrimp in responses to various shrimp pathogens.

Based on scientific consideration, the manuscript contains interesting findings and can be publishably trended in a standard journal. Generally, the manuscript is acceptably written and includes informative content that can be further applied to shrimp aquaculture. However, there are many significant concerns that the authors must pay attention to improve the quality of the manuscript. The major and minor problems are as followed;

Abstract

1) Keep consistent in indicating the gene or protein status of the MEF2 throughout the manuscript.

2) Line 28-29. The following sentence is unclear; “The results provide an important basis for future studies of the MEF2 gene in crustaceans and a clue to promote the molecular breeding of shrimp.”. The authors should clarify “…to promote the molecular breeding of shrimp”!!  

1. Introduction

Line 43. Correct “drosophila” to “Drosophila” and italicize.  

2. Results

The authors should improve the description flow by correcting many points in this part.

To increase the better flow of the manuscript, the description used for discussion, such as “suggested that, indicated that, suggesting, or any discussion contents, etc.,” must be moved to the “Discussion” section!!, throughout.  

Line 136. Clarify “…, and MEF2B has no HTURP-C domain.”.

Line 139 and 142. Correct “KD” to “kDa”, and anywhere else.

Line 143. Correct “pI” by italicizing “I” and anywhere else.

2.3. LvMEF2 gene expression profiles

Line 173-175. Description of the following information is wrong; “…and some splice variants were not expressed during the molting phase, such as MEF2-12, suggested that the different splice 174 variants of LvMEF2 played diverse roles during the molting process (Figure 4C).”. Since MEF2-12 were highly ubiquitously expressed in all molting stages.

Additionally, please remove discussion contents such as “suggested that the different splice variants of LvMEF2 played diverse roles during the molting process” from this discussion.

Fig 4A and 4F. No statistically significant differences were indicated in these two illustrations.

 Line 178. The term “hemolymph” must be adequately replaced with “hemocytes” throughout the manuscript.

Line 193. What’s the “epidermis” the authors mentioned? Is it “subcuticular epithelium”??

 Fig 5D. The authors should indicate the target parameters as mean+/-SD, and significant differences among tested groups must be shown.

2.5. Histology and bacterial agglutination experiment after LvMEF2 knockdown

 Line 247. What’s the meaning of “nuclear agglutination”? Please correctly modify. Those aggregations can be caused by hemocytes aggregating at the site observation. Higher magnification is needed.

Line 249. What’s “bacterial agglutination” mean?? The number of bacterial colonies could not appropriately indicate bacterial agglutination in shrimp, which is seriously wrong. Additionally, this experiment was improperly bad since the authors had never induced all experimental shrimp with any pathogenic bacteria (4.7), suggesting that all tested shrimp were commonly contaminated with Vibrio spp., which was not good enough for experimental conditions. Therefore, this unreliable part should be removed.

Table 2. Correct “Immunity and stress-related” to “Immunity and cellular stress-related”.

Line 32. Correct “….ligase genes Amino acids were…”.

Line 329 and information in “Table 2”. Since “immunoglobulin” has not been identified in all crustaceans or invertebrates so far, this crucial information must be carefully described. Does the current report the first identified in invertebrates?  

Since the authors used one-way ANOVA (with pos hoc, maybe) to test statistical analysis, the symbols used to indicate significant differences among all tested groups should be different letters instead of asterisks. This must be corrected in all related figures.

3. Discussion

Add the crucial information suggested above in this part correctly.

Line 374-3376. Information in these lines needs a supportive reference.

Line 385 and 431. Keep consistent in using “Oka organ” throughout the manuscript. Not lymphoid organ Oka.

Line 433. Please carefully reconsider the logic of the following information; “….and the expression of related genes in the immune pathway was up-regulated after LvMEF2 knockdown. These results suggest that MEF2 played an important role not only in muscle formation and growth but also in the immunity of shrimp.”???

4. Materials and Methods

4.1 Experimental animals

The crucial information about experimental conditions, such as water quality, must be indicated in more detail. Furthermore, the stages, size, or age of shrimp must first be stated in this part.  

4.2. Identification and analysis of MEF2 gene in L. vannamei

Line 456. Correct “for” to “from”.

4.4. RNA extraction and cDNA synthesis

Line 480. Information about “different tissues” must be first clarified in this place.

4.5. Gene expression pattern analysis

Line 493. How is “the sample of RNA interference experiments” involved in this part??

Line 497. Reference(s) of 2^delta-delta Ct must be indicated.

Line 498-500. The statistical analysis description is severely wrong; please overhaul correction this part. Basically, ANOVA cannot use to test significant differences among groups.

4.6. Double-stranded RNA synthesis and interference experiments of LvMEF2

Line 506. A full description of EGFP should be first indicated at this point.

Line 507-508. Italicize “in vitro or in vitro” throughout.

Line 518-519. It was wondering why the authors used the last abdominal segment to inject the experimental shrimp since this area is very fragile, and risky to kill the shrimp quickly.

 Line 529-530. Please clearly verify how the authors detect the expression of muscle-related genes in this part.

4.7. Tissue section and bacterial agglutination experiment

As mentioned above, the bacterial agglutination experiment is seriously not sound, has a poor design, and is unreliable; please remove it.

4.8. Transcriptome sequencing analysis of muscle samples after LvMEF2 interference

Validation analysis of transcriptomic information and qRT-PCR must have been experimented with instead.

Line 594. Please correct the awkward content of the following content; “Table S1 Primers for this manuscript”.

Line 611. The license number of this document must be declared.

Line 620. Did the current research conduct cell culture??

References

- There are some errors in this part, especially in inconsistent formats, the journal abbreviation, species names, and typos such as references 6, 8,9, 11, 16, 18, 22, 25, 28, 29, 30, 37, 38.

Author Response

The manuscript entitled “Gene Structure, Expression and Function Analysis of MEF2 in the Pacific White Shrimp Litopenaeus vannamei by Xia et al. demonstrates experimental attempts to molecularly characterize and identify gene structure, expression, and functional analyses of the myocyte enhancer factor 2 (MEF2) of Pacific white shrimp in various conditions. The results indicate that MEF2 was intensely involved in muscle formation and immune responses of shrimp in responses to various shrimp pathogens.

Based on scientific consideration, the manuscript contains interesting findings and can be publishably trended in a standard journal. Generally, the manuscript is acceptably written and includes informative content that can be further applied to shrimp aquaculture. However, there are many significant concerns that the authors must pay attention to improve the quality of the manuscript. The major and minor problems are as followed;

Abstract

1) Keep consistent in indicating the gene or protein status of the MEF2 throughout the manuscript.

Response: Thanks you for your suggestion. In this paper, according to IJMS requirements and published articles, we adopted the naming convention of italics for gene (e.g. LvMEF2) and orthographies for protein (LvMEF2); if the word "gene" is added, e.g. (LvMEF2 gene), the “LvMEF2” is not italicized.

2) Line 28-29. The following sentence is unclear; “The results provide an important basis for future studies of the MEF2 gene in crustaceans and a clue to promote the molecular breeding of shrimp.”. The authors should clarify “…to promote the molecular breeding of shrimp”!!  

Response: We appreciate the reviewer’s comment. The statement has been modified to “The results provide an important basis for future studies of the MEF2 gene and the mechanism of muscle growth and development in shrimp”.

  1. Introduction

Line 43. Correct “drosophila” to “Drosophila” and italicize.

Response: We have modified "drosophila" to "Drosophila" and italicize it. Thank you very much.

  1. Results

The authors should improve the description flow by correcting many points in this part.

To increase the better flow of the manuscript, the description used for discussion, such as “suggested that, indicated that, suggesting, or any discussion contents, etc.,” must be moved to the “Discussion” section!!, throughout.  

Response: Thanks for your valuable comments, and the relevant content in the manuscript has been rewritten or transferred to the discussion section.

Line 136. Clarify “…, and MEF2B has no HTURP-C domain.”.

Response: Thank you very much for your suggestion. We have added a note in Figure 2, and the HTURP-C domain is not present in human MEF2B, we have also modified the structure of the figure from a grey box to a black line to indicate the absence of the HTURP-C domain in human MEF2B.

Line 139 and 142. Correct “KD” to “kDa”, and anywhere else.

Response: We have modified “KD” to “kDa” in the manuscript.

Line 143. Correct “pI” by italicizing “I” and anywhere else.

Response: We accept reviewer’s suggestion and modify it in the manuscript.

2.3. LvMEF2 gene expression profiles

Line 173-175. Description of the following information is wrong; “…and some splice variants were not expressed during the molting phase, such as MEF2-12, suggested that the different splice 174 variants of LvMEF2 played diverse roles during the molting process (Figure 4C).”. Since MEF2-12 were highly ubiquitously expressed in all molting stages.

Additionally, please remove discussion contents such as “suggested that the different splice variants of LvMEF2 played diverse roles during the molting process” from this discussion.

Response: We are sorry for the confusion caused by the fact that the serial numbers of the splice variants in the figure were updated during the previous revisions, and that a small part of the corresponding numbers in the text were not updated. Now, the text and the corresponding discussion section have been modified. We appreciate the reviewer’s comment.

Fig 4A and 4F. No statistically significant differences were indicated in these two illustrations.

Response: Thank you for your suggestion. The significance has been added to Figure 4A and Figure 4F and illustrated in the figure legend.

Line 178. The term “hemolymph” must be adequately replaced with “hemocytes” throughout the manuscript.

Response: The word "hemolymph" has been replaced with "hemocytes" throughout the manuscript.

Line 193. What’s the “epidermis” the authors mentioned? Is it “subcuticular epithelium”??

Response: Thank you for your question, I would like to explain that shrimp are covered by a supporting and protective chitinous carapace called the exoskeleton, and an inner layer of elastic, exoskeleton-secreting epidermal cells. In our sample, the epidermis was scraped from the inner layer of the chitinous shell of the cephalothorax of the shrimp L. vannamei, the term "epidermis" for the tissue used here. The literature of published about tissue distribution shows that the term "epidermis" is used.

[1] Wang L, Lu KC, Chen GL, Li M, Zhang CZ, Chen YH. A Litopenaeus vannamei TRIM32 gene is involved in oxidative stress response and innate immunity. Fish Shellfish Immunol. 2020 Dec;107(Pt B):547-555. doi: 10.1016/j.fsi.2020.11.002. Epub 2020 Nov 6. PMID: 33161091.

[2] Yang F, Li X, Li S, Xiang J, Li F. A novel cuticle protein involved in WSSV infection to the Pacific white shrimp Litopenaeus vannamei. Dev Comp Immunol. 2020 Jan; 102:103491. doi: 10.1016/j.dci.2019.103491. Epub 2019 Sep 5. PMID: 31494218.

[3] Lian YY, He HH, Zhang CZ, Li XC, Chen YH. Functional characterization of a matrix metalloproteinase 2 gene in Litopenaeus vannamei. Fish Shellfish Immunol. 2019 Jan; 84:404-413. doi: 10.1016/j.fsi.2018.10.021. Epub 2018 Oct 11. PMID: 30316944.

Fig 5D. The authors should indicate the target parameters as mean+/-SD, and significant differences among tested groups must be shown.

Response: Thank you for your suggestion. In the revised version, we use the line graph of cumulative death to replace that of molting, which can more clearly show the effect of LvMEF2 RNA interference on the experimental shrimp.

2.5. Histology and bacterial agglutination experiment after LvMEF2 knockdown

Line 247. What’s the meaning of “nuclear agglutination”? Please correctly modify. Those aggregations can be caused by hemocytes aggregating at the site observation. Higher magnification is needed.

Response: Thank you for your question. During the RNA interference experiment, we found that the abdominal muscle in the experimental group shrimp was softer than that of control group, so we conducting tissue sectioning experiments to observe the differences in muscle tissue between the experimental group and the treatment group. Through observation, we found that the muscle tissue space was enlarged and there was nucleus agglutination, which may be blood cells, indicating that the whole structure of the muscle tissue was affected, which may be the reason for the soft muscle. Deeper changes in muscle tissue structure are difficult to observe under the light microscope, and may require electron microscopy or histochemistry. We will carry out research in this field in the future

Line 249. What’s “bacterial agglutination” mean?? The number of bacterial colonies could not appropriately indicate bacterial agglutination in shrimp, which is seriously wrong. Additionally, this experiment was improperly bad since the authors had never induced all experimental shrimp with any pathogenic bacteria (4.7), suggesting that all tested shrimp were commonly contaminated with Vibrio spp., which was not good enough for experimental conditions. Therefore, this unreliable part should be removed.

Response: Thank you very much for your suggestion. We have had a careful discussion on this question, and it may be more appropriate to change the name of the experiment to total viable bacteria count after LvMEF2 knockdown. In the RNA interference experiment, we set up three groups of shrimp (72 individuals/group), named the dsMEF2 group, dsEGFP group and PBS group. We did not use any bacteria or viruses during the whole RNA interference experiment, and we changed seawater every day. However, only the shrimp in the dsMEF2 group died in the later stage of the experiment. The phenomenon exceeded our expectations and excited our curiosity. Since shrimp contains conditional pathogens in hepatopancreas [1-3], we also found that there are many studies in other animals that show LvMEF2 is related to immunity. Therefore, we suspect that the knockdown of the LvMEF2 gene affects the immune system of shrimp, leading to the destruction of the conditional pathogens homeostasis in shrimp, then resulting in the death of the shrimp. Therefore, we conducted a total viable bacteria count experiment and found that the total viable bacteria in the dsMEF2 group hepatopancreas was much higher than that in the control group without any external bacteria introduction[4-5]. Therefore, this experiment confirmed that the interference of LvMEF2 gene affected the immune system of shrimp, leading to an increase in the number of conditional pathogens. The results rule out other possible causes of death, such as hypoxia etc. Since our conclusion is that MEF2 gene affects the growth, metabolism and immunity of shrimp, and this part of the experiment is important supporting evidence, we have not removed this part of the content.

[1] Ye, R., et al., Separation and identification of endogenous dominant spoilage bacteria from Litopenaeus vannamei. Journal of Fisheries of China, 2013. 37(9): p. 1425-1430.

[2] Dai L, Xiong Z, Hou D, Wang Y, Li T, Long X, Chen H, Sun C. Pathogenicity and transcriptome analysis of a strain of Vibrio owensii in Fenneropenaeus merguiensis. Fish Shellfish Immunol. 2022 Nov;130: 194-205. doi: 10.1016/j.fsi.2022.09.008. Epub 2022 Sep 7. PMID: 36087819.

[3] Chen X, Zeng D, Chen X, Xie D, Zhao Y, Yang C, Li Y, Ma N, Li M, Yang Q, Liao Z, Wang H. Transcriptome analysis of Litopenaeus vannamei in response to white spot syndrome virus infection. PLoS One. 2013 Aug 26;8(8):e73218. doi: 10.1371/journal.pone.0073218. PMID: 23991181; PMCID: PMC3753264.

[4] Sun M, Li S, Lv X, Xiang J, Lu Y, Li F. A Lymphoid Organ Specific Anti-Lipopolysaccharide Factor from Litopenaeus vannamei Exhibits Strong Antimicrobial Activities. Mar Drugs. 2021 Apr 28;19(5):250. doi: 10.3390/md19050250. PMID: 33925052; PMCID: PMC8145222.

[5] Lv X, Li S, Yu Y, Zhang X, Li F. Characterization of a gill-abundant crustin with microbiota modulating function in Litopenaeus vannamei. Fish Shellfish Immunol. 2020 Oct;105:393-404. doi: 10.1016/j.fsi.2020.07.014. Epub 2020 Jul 21. PMID: 32702477.

Table 2. Correct “Immunity and stress-related” to “Immunity and cellular stress-related”.

Response: We have modified the word as your suggestion.

Line 32. Correct “….ligase genes Amino acids were…”.

Response: Thanks, we have deleted “Amino acids” in the revised manuscript.

Line 329 and information in “Table 2”. Since “immunoglobulin” has not been identified in all crustaceans or invertebrates so far, this crucial information must be carefully described. Does the current report the first identified in invertebrates?  

Response: Thanks for your question. We have had a careful discussion and checked these genes. First, the proteins mentioned in the table are not immunoglobulin proteins, but they all contain immunoglobulin-I-set domains. We translated the transcripts of the six genes into amino acid sequences and performed protein domain prediction in the SMART website. They were found to have either IGc2 domains or IG domains, and some had both (as shown below). These domains all have immune-related functions. So we think they can be classified as immune-related proteins. In fact, their specific function, and the relationship with Immunoglobulins remain to be studied, which may be clarified in the future.

> LVAN16387

MLVVAVFLTAMLAVAGEDVEAVEAVEAVDAFEVLDPSTLVGQLTVLQLKELIAETMAQALVPLHHNCTDYSETGDCGLVVENGLCGAEEFYARYCCRSCTLAKQIPTYGPHLSLTAKAPISINVTTDLVKYPRGSNVTITCEATGYPTPEVVWFRNYNEISSIGEYEEDTEVLGGILLKTIRSNLIIDNYSVDYSHFQCMAINSEGLAKSSITIYIEDELSVVLLPEVPIMEAKNEVVLDCVATGADVNEVIWFREVELIHNSNEYRIIERRSFEDGLTKIHSKLTILRHKTEDSGSLFTCSAIDRKGGNKEKKACYEESHRCEFKQEYIYRYCCKTCTLMGVLPTYGPHLRGDAEAPLEVSIQTNSTVVPYGGDAELVCMTSGFPGRQSVMWLKKDELIRESNNIHIKHNSACSPLLCEVSASLLIKGIKISDTGKYICRTSNVVGMKDALVLLRSMEDGEIYIDIDY*

 https://smart.embl.de/

IGc2 domain(Interpro IPR003598):

SMART ACC:  SM000408

Definition:     Immunoglobulin C-2 Type

Interpro abstract (IPR003598):   

The basic structure of immunoglobulin (Ig) molecules is a tetramer of two light chains and two heavy chains linked by disulphide bonds. There are two types of light chains: kappa and lambda, each composed of a constant domain (CL) and a variable domain (VL). There are five types of heavy chains: alpha, delta, epsilon, gamma and mu, all consisting of a variable domain (VH) and three (in alpha, delta and gamma) or four (in epsilon and mu) constant domains (CH1 to CH4). Ig molecules are highly modular proteins, in which the variable and constant domains have clear, conserved sequence patterns. The domains in Ig and Ig-like molecules are grouped into four types: V-set (variable; IPR013106 ), C1-set (constant-1; IPR003597 ), C2-set (constant-2; IPR008424 ) and I-set (intermediate; IPR013098 ) [ (PUBMED:9417933) ]. Structural studies have shown that these domains share a common core Greek-key beta-sandwich structure, with the types differing in the number of strands in the beta-sheets as well as in their sequence patterns [ (PUBMED:15327963) (PUBMED:11377196) ].

Immunoglobulin-like domains that are related in both sequence and structure can be found in several diverse protein families. Ig-like domains are involved in a variety of functions, including cell-cell recognition, cell-surface receptors, muscle structure and the immune system [ (PUBMED:10698639) ].

This entry represents a subtype of the immunoglobulin domain, and is found in a diverse range of protein families that includes glycoproteins, fibroblast growth factor receptors, vascular endothelial growth factor receptors, interleukin-6 receptor, and neural cell adhesion molecules. It also includes proteins that are classified as unassigned proteinase inhibitors belonging to MEROPS inhibitor families I2, I17 and I43 [ (PUBMED:14705960) ].

IG domain(Interpro IPR003599):  

SMART ACC:  SM000409

Definition:     Immunoglobulin

Interpro abstract (IPR003599):   

The basic structure of immunoglobulin (Ig) molecules is a tetramer of two light chains and two heavy chains linked by disulphide bonds. There are two types of light chains: kappa and lambda, each composed of a constant domain (CL) and a variable domain (VL). There are five types of heavy chains: alpha, delta, epsilon, gamma and mu, all consisting of a variable domain (VH) and three (in alpha, delta and gamma) or four (in epsilon and mu) constant domains (CH1 to CH4). Ig molecules are highly modular proteins, in which the variable and constant domains have clear, conserved sequence patterns. The domains in Ig and Ig-like molecules are grouped into four types: V-set (variable; IPR013106 ), C1-set (constant-1; IPR003597 ), C2-set (constant-2; IPR008424 ) and I-set (intermediate; IPR013098 ) [ (PUBMED:9417933) ]. Structural studies have shown that these domains share a common core Greek-key beta-sandwich structure, with the types differing in the number of strands in the beta-sheets as well as in their sequence patterns [ (PUBMED:15327963) (PUBMED:11377196) ].

Immunoglobulin-like domains that are related in both sequence and structure can be found in several diverse protein families. Ig-like domains are involved in a variety of functions, including cell-cell recognition, cell-surface receptors, muscle structure and the immune system [ (PUBMED:10698639) ].

Since the authors used one-way ANOVA (with pos hoc, maybe) to test statistical analysis, the symbols used to indicate significant differences among all tested groups should be different letters instead of asterisks. This must be corrected in all related figures.

Response: Thanks for your suggestion. The detection method used in this paper is the t-test. Due to our negligence, the significance detection method has not been modified in the material method. All the contents of this part of the text have been revised in manuscript and supplemented materials now.

  1. Discussion

Add the crucial information suggested above in this part correctly.

Line 374-376. Information in these lines needs a supportive reference.

Response: Thanks for your advices. The supportive references had been added in the revised manuscript.

[1] Moretti, I., et al., MRF4 negatively regulates adult skeletal muscle growth by repressing MEF2 activity. Nature Communications, 2016. 7: p. 12397.

[2] Bird, L., Immunometabolism: Mef2 in sickness and in health. Nature Reviews Immunology, 2013. 13(12): p. 845.

Line 385 and 431. Keep consistent in using “Oka organ” throughout the manuscript. Not lymphoid organ Oka.

Response: Modified. Thanks for your suggestion.

Line 433. Please carefully reconsider the logic of the following information; “….and the expression of related genes in the immune pathway was up-regulated after LvMEF2 knockdown. These results suggest that MEF2 played an important role not only in muscle formation and growth but also in the immunity of shrimp.”???

Response: We appreciate the reviewer’s comment. The phrase “play an important role” is not rigorous here. This has been amended to “These results suggest that MEF2 had an important influence not only in muscle formation and growth, but also in the immunity of shrimp”.

  1. Materials and Methods

4.1 Experimental animals

The crucial information about experimental conditions, such as water quality, must be indicated in more detail. Furthermore, the stages, size, or age of shrimp must first be stated in this part.  

Response: Thanks for raising this important point via this comment. We revised it as follows “The shrimp used in this experiment had a body length of 7.5±0.5cm and a body weight of 4.5 ±0.5g. Before the experiment, the shrimp were cultured in the breeding tank for a week, and the temperature of the aerated seawater was maintained at 25 ± 1℃ with the salinity of 30‰ and pH 7.5 ± 0.1”.

4.2. Identification and analysis of MEF2 gene in L. vannamei

Line 456. Correct “for” to “from”.

Response: Done.

4.4. RNA extraction and cDNA synthesis

Line 480. Information about “different tissues” must be first clarified in this place.

Response: Thanks for your comment. Information about “different tissues” has been deleted in the revised manuscript.

4.5. Gene expression pattern analysis

Line 493. How is “the sample of RNA interference experiments” involved in this part??

Response: We are sorry for the misleading, here, we refer to the q-PCR detection of the RNA interference experimental samples in the later. Considering that section 4.6 and 4.8 also mentioned q-PCR of interference experiment, here we deleted “the sample of RNA interference experiments” in the revised manuscript.

Line 497. Reference(s) of 2^delta-delta Ct must be indicated.

Response:Thanks for your comment. Reference(s) about “2^delta-delta Ct” has been added in the revised manuscript.

[1] Livak, K.J. and T.D. Schmittgen, Analysis of Relative Gene Expression Data Using Real-Time Quantitative PCR and the 2−ΔΔCT Method. Methods, 2001. 25(4): p. 402-408.

Line 498-500. The statistical analysis description is severely wrong; please overhaul correction this part. Basically, ANOVA cannot use to test significant differences among groups.

Response: Thanks for your comment. The significance analysis method used in the analysis of gene expression patterns was one-way ANOVA, and the significance analysis method used in the subsequent interference experiment was the t-test. We have also made corresponding clarifications.

4.6. Double-stranded RNA synthesis and interference experiments of LvMEF2

Line 506. A full description of EGFP should be first indicated at this point.

Line 507-508. Italicize “in vitro or in vitro” throughout.

Response: Thanks for your suggestion. The relevant format has been modified in the revised manuscript.

Line 518-519. It was wondering why the authors used the last abdominal segment to inject the experimental shrimp since this area is very fragile, and risky to kill the shrimp quickly.

Response: We appreciate the reviewer’s comment. We are very sorry for our incorrect statement, the dsRNA was injected intramuscularly at the fourth abdominal segment.

Line 529-530. Please clearly verify how the authors detect the expression of muscle-related genes in this part.

Response: Thanks for your comment. It has been modified as “…. the expression of LvMEF2 and muscle-related genes was detected by quantitative fluorescence PCR (qPCR). The data were analyzed by the t-test using GraphPad Prism soft-ware. p<0.05 was marked as a single asterisk.”

4.7. Tissue section and bacterial agglutination experiment

As mentioned above, the bacterial agglutination experiment is seriously not sound, has a poor design, and is unreliable; please remove it.

Response: Thanks for your suggestion. The experiment was explained in detail in the results section, where the name and procedure of the experiment were revised.

4.8. Transcriptome sequencing analysis of muscle samples after LvMEF2 interference

Validation analysis of transcriptomic information and qRT-PCR must have been experimented with instead.

Response: Thanks for your valuable advice. In the section of “4.8. Transcriptome sequencing analysis of muscle samples after LvMEF2 interference”, we verified the reliability and accuracy of transcriptome sequencing and analysis. Ten DEGs were selected to detect the expression by RT-qPCR, the result confirmed the accuracy of RNA-Seq results (Figure S4). For the follow-up verification of other analysis results of the transcriptome, we will design relevant strict experiments to verify.

Line 594. Please correct the awkward content of the following content; “Table S1 Primers for this manuscript”.

Response: We appreciate the reviewer’s comment. It has been modified as "Table S1 Primers used in this research".

Line 611. The license number of this document must be declared.

Response: We have added the license number in the revised manuscript.

Line 620. Did the current research conduct cell culture??

Response: We appreciate the reviewer’s comment. We tried some cell work to verify the regulation of LvMEF2 genes, but no obvious progress has been made now. Since these results are not included in this article, the acknowledgments regarding the relevant persons who helped in the cell culture process have been deleted.

References

- There are some errors in this part, especially in inconsistent formats, the journal abbreviation, species names, and typos such as references 6, 8,9, 11, 16, 18, 22, 25, 28, 29, 30, 37, 38.

Response: Thanks for your suggestion, we have revised these references accordingly.

Reviewer 4 Report

The transcription factor Myocyte Enhancer Factor 2 (MEF2) plays a key role in muscle development, differentiation and embryonic development. MEF2 gene from Litopenaeus vannamei was characterized in the current study by Xia et al. The expression of MEF2 was analysed in different tissues. Using transcriptome sequencing, alternative splicing variants of MEF2 were identified. Silencing of the MEF2 gene was carried out by RNAi and the effect on growth, survival etc was analysed. Transcriptomics was used to identify global overall gene expression profiling after RNAi-mediated MEF2 gene silencing. In this study, the authors have used methods that are generally accepted for this type of research, the paper has been well-written, and data has been analyzed using appropriate statistical procedures.

A few suggestions are provided below to help improve the manuscript.

1. Cloning the MEF2 – Most of the sequence data obtained by the authors are based on transcriptomic sequencing data using short read technology. It would have been better if the authors had amplified and cloned at least one (The canonical isoform ) Litopenaeus MEF2 gene (cDNA fragment encoding the protein coding region/ORF) and  confirmed that it matches with transcriptomic data.

2. Lines 495-497 – Section 4. 5  - Details of qPCR as per MIQE guidelines should be provided here:  ( with annealing /extension temperature and the name of the primers used need to be  clearly mentioned here). Also mention the details of RT-qPCR ( qPCR Mix used, type (SYBR Green), primer/reagent concentrations, qPCR equipment as per the Minimum Information for Publication of Quantitative Real-Time PCR Experiments (MIQE) guidelines

Table S1 – Line 3 – Size of the PCR product of 18-SF/R primers Please correct the unit to bp.

Table S1 – The Table S1 lists different primer pairs used for different purposes ( gene expression, RNAi etc). It would be helpful if you can add another column on the right side of table indicating the purpose of each primer pair.

Author Response

Thank you for your comment concerning our manuscript. I accept your valuable suggestions and make a one-to-one response in letter.

Round 2

Reviewer 3 Report

Reviewers' Comments to Authors:

The revised manuscript entitled “Gene Structure, Expression and Function Analysis of MEF2 in the Pacific White Shrimp Litopenaeus vannamei by Xia et al. demonstrates experimental attempts to molecularly characterize and identify gene structure, expression, and functional analyses of the myocyte enhancer factor 2 (MEF2) of Pacific white shrimp in various conditions. The results indicate that MEF2 was intensely involved in muscle formation and immune responses of shrimp in responses to various shrimp pathogens.

The revised manuscript has almost responded all previously raised recommendations and great enough to be published in the IJMS. By the way, some minor corrections are still needed to improve the quality of the manuscript. These included,

1) The gene and protein statuses of MEF2 must be corrected throughout the manuscript.  

2) Line 570. Correct “TCBS agar media” to “TCBS agar”.

3) References

There are some errors in this part, especially in inconsistent and error formats, the journal abbreviation, species names, and typos. Please take a look on the journal guideline.

Author Response

The revised manuscript entitled “Gene Structure, Expression and Function Analysis of MEF2 in the Pacific White Shrimp Litopenaeus vannamei by Xia et al. demonstrates experimental attempts to molecularly characterize and identify gene structure, expression, and functional analyses of the myocyte enhancer factor 2 (MEF2) of Pacific white shrimp in various conditions. The results indicate that MEF2 was intensely involved in muscle formation and immune responses of shrimp in responses to various shrimp pathogens.

The revised manuscript has almost responded all previously raised recommendations and great enough to be published in the IJMS. By the way, some minor corrections are still needed to improve the quality of the manuscript. These included,

Question1: The gene and protein statuses of MEF2 must be corrected throughout the manuscript.  

Response: Thank you very much for your comment. We are sorry that we didn't understand what the mean by "Keep consistent in indicating the gene or protein status of the MEF2" very well. Does either the gene or the protein was shown to be changed into one status? Here we modified to show all genes in italics and all proteins in orthographies according to the articles published on IJMS. We hope that these modifications meet the requirements. If there are any problems, we would appreciate your suggestions very much.

Question 2: Line 570. Correct “TCBS agar media” to “TCBS agar”.

Response: We have modified "TCBS agar media" to "TCBS agar". Thank you very much.

Question 3: References

There are some errors in this part, especially in inconsistent and error formats, the journal abbreviation, species names, and typos. Please take a look on the journal guideline.

Response: Thank you for your suggestion, we have revised these references accordingly.